# Study of a Control Algorithm with the Disturbance of Massive Discharge on an Open Channel

**Jian Shen [1], Bo Kang [2,\*], Yuezan Tao [1], Fei Lin [3] and Xuegong Song [4]**

1 School of Civil Engineering, Hefei University of Technology, Hefei 230009, China
2 School of Resource and Environmental Engineering, Hefei University of Technology, Hefei 230009, China
3 School of Resources and Environmental Engineering, Anhui University, Hefei 230601, China
4 Shandong Province Water Transfer Project Operation and Maintenance Center Shouguang Management Station, Shouguang 262700, China
\* Correspondence: kangbo@hfut.edu.cn; Tel.: +86-1896-3794-495

**Abstract:** The diversion of large flows of water in open channels, which exceed safe water levels, into water storage systems is the focus of this paper. We focused upon the middle route of the South-to-North Water Transfer Project and the ancient canal gate 4 drainage pool. We estimated the water storage quantity and the amount of compensation this offered. We used the improved PID algorithm to the study 20 disturbance flow water pipe heads. The results show that the storage compensation algorithm can suppress the fluctuation of water level to a certain extent, but, for the disturbance caused by large flows, the maximum fluctuation of water level in front of Shahei sluice gate that the storage compensation control algorithm is capable of is more than 0.3 m, which is much larger than the water level constraint of the middle line channel pool, where the rate of the drop in water level should not exceed 0.15 m per hour. However, in the case of large flow disturbance, the variation of water level in front of all the control gates in the study area is not more than 0.1 m, which meets the severe water level constraint of the middle line project, and the water level control effect is good, which protects the operation safety of the canal pool.

**Keywords:** water diversion project; the south-to-north water diversion project; storage compensation algorithm; improved PID control algorithm; coupled control model





## 1. Introduction

Chinese water distribution is more concentrated in the east and less in the west, and more in the south and less in the north. With rapid economic and social development, the contradiction between the supply and demand of water resources in China is becoming more prominent [1]. To optimize the allocation of water resources in China and alleviate the contradiction between the supply and demand of water resources, the building of large-scale water transfer projects is the most effective and direct method. The middle line project of the South-to-North Water Diversion Project is a significant water transfer project, which aims to alleviate the shortage of water resources in the north, optimizing water resource allocation, improving the ecological environment, and driving economic growth. Compared with other water transfer projects, this project has a large water flow, long water lines, and numerous hydraulic structures; so, there are many challenges in the safe operation of this water transfer project [2,3].

The middle line of the South-to-North Water Diversion Project operates at a constant water level in front of the gate and only requires a small range of fluctuations in the water level, and the operating conditions are harsh [4]. Therefore, when there is a demand for water downstream, the safe operation of the channel is an urgent problem that needs to be solved. In addition, it can be found from the studies of some authors that the flow pattern of the flow velocity is not fixed during the process of water diversion. Lama, G.F.C. et al. [5] used the first-order and second-moment statistical method to prove the uneven distribution

of the flow velocity on the section surface. Mohammad Amir Khan et al. [6] used Doppler acoustic velocimetry to measure and calculate the three-dimensional flow field distribution of the channel. All these results greatly proved that the control of water levels control in open channels is extremely complex to study. Many scholars around the world have studied the constant water level operation in front of the central line project gate. Yang Kailin [7] further studied the way that the sluice gate joint operation had to be controlled to maintain the constant water level in front of the sluice gate. Fang Shenguang et al. [8] also studied the way that the sluice gate joint operation had to be controlled to maintain the constant water level in front of the sluice gate. Cui Wei et al. [9] established a channel space controller and achieved good results in the simulation process by coupling the feedforward control of the storage compensation and the feedback control of the proportional-integral-derivative (PID). Wu Huiming, Lei Xiaohui et al. [10] compared the active flow compensation with the downstream constant water level control to verify the rationality of the constant water level control algorithm in front of the gate. The middle line project control problem can be summarized as a large time delay and strong coupling. Yan Peiru et al. [11] proposed a feedforward control method based on hydraulic response time, which was verified by flume experiment to ensure the stability of the water level in front of the pump station in the process of flow switching. Gu Weiwei et al. [12] proposed the coupling of the extended state observer with the traditional model prediction algorithm. The experimental results show that the controller has high performance and strong robustness to various model uncertainties. Maryam Jaban Salehi et al. [13] proposed a comparison of real-time control effects based on MPC and particle swarm optimization (PSO) algorithm in large open channels. The MPC model has a higher effect on multi-variable optimization than the PSO optimization algorithm, but the solution of the MPC algorithm model is more complex and requires higher accuracy. In comparison, the PSO control algorithm is simpler and easier to implement. MPC is indeed superior to other algorithms in terms of precise control, but the midline project adopts the water level interval control. When the water distribution plan is known, the feedforward control coupled PID control algorithm can already meet the effect of interval control. Moreover, the governing equation of the MPC model is more complex, the code is more difficult to implement, the solution accuracy is higher, and the cost of implementation in the middle line of the South-to-North Water transfer is larger, so this paper adopts the feedforward control coupled PID control algorithm as the control method of the system.

The operation control problem of the middle route of the South-to-North Water transfer project can be summarized as large time delay and strong coupling [14]. When downstream water demands increase, water flow downstream usually takes a few days; in order to meet the demand downstream, the gate requires the time of a few days for feedforward control decision-making and execution [15,16], but when a channel undergoes the process of water pool level changes, almost all of the drainage basin are impacted by the influence of water level change, and the safe operation of the channel becomes challenging when there are significant changes unfavorable to the midline; this problem is even more pronounced in the case of large flows of water [17]. This paper analyzes the case of large flow disturbance, and compares the control effect of two algorithms, the storage compensation algorithm and the storage compensation coupled improved PID control algorithm, respectively [18].

## 2. Construction of Hydraulic Simulation Model

### 2.1. Governing Equation

The foundation of the hydraulic simulation model is the establishment of a hydrodynamic model and the building of a one-dimensional non-constant hydrodynamic model to simulate the hydraulic system of the entire system [19]. The basic equations for solving the one-dimensional hydrodynamic model are the Saint-Venant equations:

Continuity equation:

$$B\frac{\partial Z}{\partial t} + \frac{\partial Q}{\partial x} = q \tag{1}$$

Momentum equation:

$$\frac{\partial}{\partial t}\left(\frac{Q}{A}\right)+\frac{\partial}{\partial x}\left(\frac{Q^2}{2A^2}\right)+g\frac{\partial h}{\partial x}+g(S_f-S_0)=0 \tag{2}$$

In the Equation, $Z$ is the water level of the section in m; $B$ is the width of the water-passing section in m; $h$ is the water depth of the section in m; $Q$ is the flow rate of the section in m$^3$/s; $x$ is the space coordinate, $t$ is the time coordinate, $q$ is the inflow from the side of the channel per unit length, in m$^3$/s; $A$ is the area of the water-passing section in m$^2$; $g$ is the acceleration due to gravity in m/s$^2$; S$_0$ is channel specific drop, $R$ is the hydraulic gradient, in m; and $S_f$ is the friction-resistance ratio drop. The Equation is as follows:

$$S_f=\frac{Q^2n^2}{A^2R^{4/3}} \tag{3}$$

In the Equation, $n$ is the roughness, in s/m$^{1/3}$.

The Saint-Venant equations are first-order linear hyperbolic partial differential equations, and the analytical solution of the equation cannot be obtained accurately through calculation. In this research, the Preissmann four-point method (Figure 1) with a fast convergence speed is used for discretization, and a grid equation system is established [20]. Combined with the upstream and downstream boundary conditions, a closed non-linear equation system is obtained, and the chase method is used to solve the system. The hydraulic structures inside the canal pool, such as the gate, the water outlet, and the inverted siphon, are generalized according to the inner boundary. For details, please refer to the research results of Kong Linzhong et al. [21].

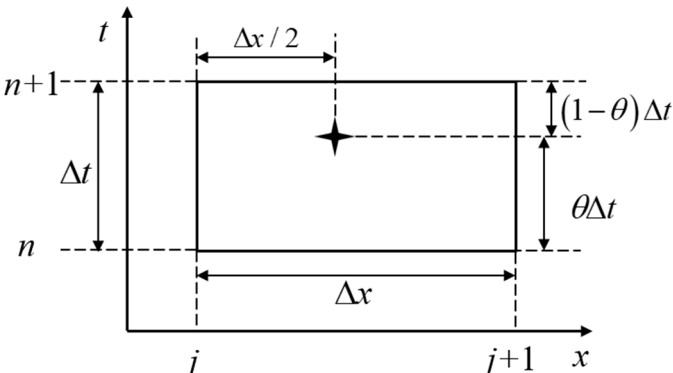

**Figure 1.** Preissmann four-point weighted implicit difference scheme.

### 2.2. Inner Boundary Handling

The middle line project is a large-scale water transfer project formed by several channels and pools in series. The middle of each channel pool is connected by a control gate (Figure 2); so, the control gate can be summarized as the inner boundary of the channel pool [22]. The over-gate flow at the inner boundary can be derived from the gate outflow Equation:

$$Q_g=C_dbu\sqrt{\Delta h} \tag{4}$$

In the Equation, $Q_g$ is the gate flow rate, in m$^3$/s; $C_d$ is the gate flow coefficient, $u$ is the gate opening, $b$ is the bottom width of the gate, in m; and $\Delta h$ is the difference between the water level before and after the control gate, in m.

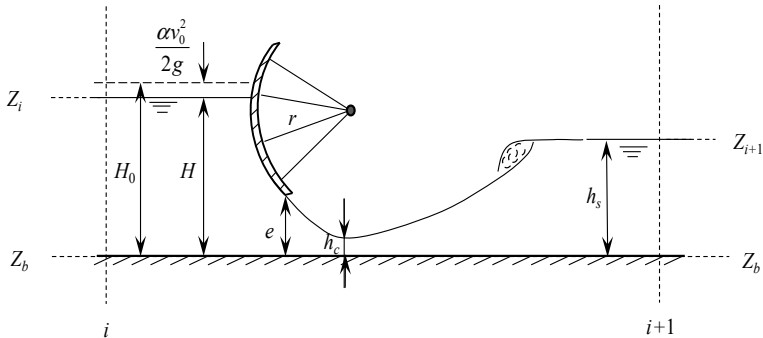

**Figure 2.** Schematic diagram of control gate overcurrent.

*2.3. Water Divider Generalization*

The function of the water distribution port (Figure 3) is to provide enough water to the water-receiving area. Usually, the water distribution port is also the main factor of the disturbance of the water level change; so, the water distribution port is the disturbance of the system [23]. The flow change of the water outlet satisfies the water balance equation:

$$Q_j=Q_{j+1}+Q_n \tag{5}$$

In the Equation, $Q_j$ represents the flow before water separation, in $m^3/s$; $Q_{j+1}$ represents the flow after water separation, in $m^3/s$; and $Q_n$ represents the water separation flow at the water outlet, in $m^3/s$. Due to the small distance between the water distribution openings, it is generally considered that the water level does not change before and after the water distribution:

$$Z_j=Z_{j+1} \tag{6}$$

In the equation, $Z_j$ represents the water level value of the channel section before water diversion, in m, and $Z_{j+1}$ represents the water level value of the channel section after water diversion, in m.

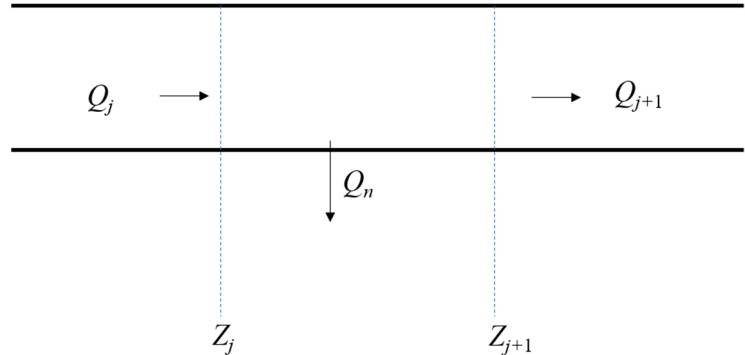

**Figure 3.** Bleeder schematic.

## 3. Control Algorithm in the Case of Large Flow Diversion Disturbance

This section may be divided by subheadings. It should provide a concise and precise description of the experimental results, their interpretation, as well as the experimental conclusions that can be drawn.

*3.1. Stepwise Storage Compensation Algorithm*
Traditional Storage Compensation Algorithm

The storage compensation algorithm is a feedforward control method that draws up control rules for gates along the line based on downstream water demand requirements.

The storage compensation method [24] has the advantages of simplicity and good stability, and this method has been applied to the automatic control research of the Salt River project in the United States [25].

The principle of the storage compensation algorithm for a channel with a single canal pool and a single water outlet is shown (Figure 4). It is assumed that the diversion head suddenly changes by $\Delta q_d$ at the downstream moment $t_d$, and the corresponding channel storage volume changes by $\Delta V$ at the same time, as shown in the shaded part of Figure 4. To keep the water level in the canal pool stable, it is necessary to increase the upstream inflow by $\Delta Q_1$ at time $t_1$ in advance. In the figure, $\Delta \tau$ is the propagation time of the disturbance wave (delay time). Finally, in order to balance the inflow and outflow, it is necessary to reduce the inflow by $\Delta Q_2$ at $t_d$.

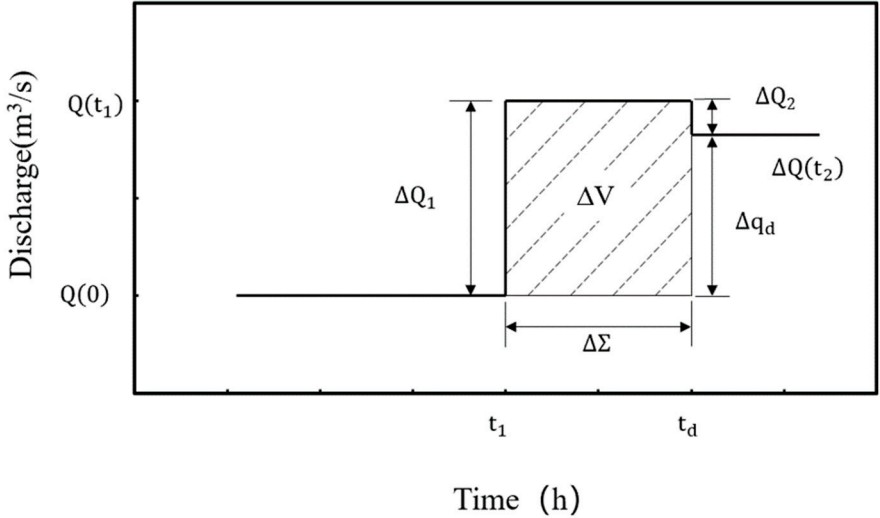

**Figure 4.** Schematic diagram of storage compensation algorithm.

In the case of multi-channel and multi-distribution, the calculation equation of a single canal pond is referred to. According to the water diversion demand of each canal pond, the feedforward strategy of each canal pond is formulated, and through the accumulation method, the control strategy of the canal head can be obtained by adjusting the gate inflow of the canal head to achieve the control requirements, as follows:

$$\Delta \tau = t_d - t_1 \tag{7}$$

$$\Delta Q_1 = \frac{\Delta V}{\Delta \tau} \tag{8}$$

$$\Delta Q_2 = \Delta Q_1 - \Delta q_d \tag{9}$$

$$\Delta Q_{1,m,i} = \sum_{k=m}^{N} \Delta V_{k,i} / \sum_{k=m}^{N} \Delta \tau_{k,i} \tag{10}$$

$$\sum_{k=m}^{N} \Delta \tau_{k,i} = t_{di} - t_{1,m,i} \tag{11}$$

$$\Delta Q_{2,m,i} = \Delta q_i - \Delta Q_{1,m,i} \tag{12}$$

In the equation, $N$ is the channel number where the water demand occurs, $\Delta Q_{1,m,i}$ is the first inflow adjustment made by the water demand $i$ to the $m$th canal pool, and $\Delta Q_{2,m,i}$ is the second inflow adjustment $i$ made by the water demand to the mth canal pool. $t_{1,m,i}$ represents the time at which the $\Delta Q_{1,m,i}$ adjustment is performed, and $t_{di}$ represents the time at which the $\Delta Q_{2,m,i}$ adjustment is performed. $\Delta \tau_{k,i}$ represents the lag time of the $k$th canal pond affected by the water demand $i$ of $N$ canal ponds.

For series-connected pools with a high water demand, when the water distribution of the canals and pools is too large, the conventional storage compensation algorithm is used.

There are serious water supply problems, such as flooding. Therefore, a distributed storage compensation algorithm is proposed for large flow disturbances. Due to the operation of the middle line, in order to maintain the safety of the operation of the middle line project, there is a strict limit on the amplitude of the water level. The rate of water level drop is generally 0.15 m/h, and it can drop by at most 0.30 m in 1 day [26,27]. The so-called distributed storage compensation means that with the constraint of the water level fluctuation, the original $\Delta Q$ of one-time regulation is divided into several steps to achieve the regulation, thereby reducing the oscillation of the water level without generating the callback flow and reducing the amount of gate regulation, as shown in Figure 5.

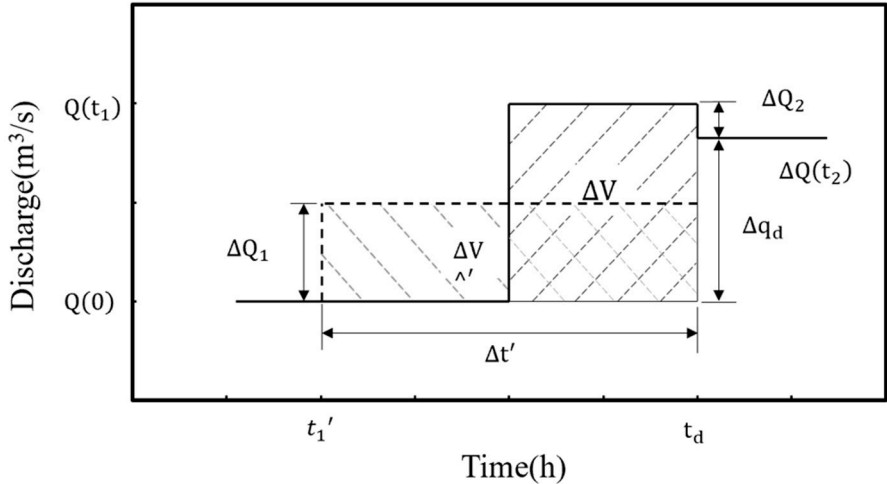

**Figure 5.** Schematic diagram of distributed storage compensation algorithm.

In the case of multi-canal and multi-demand conditions, the compensation strategy of each canal and pond is determined based on the water distribution of the canal and pond and the constraints of the water level. The occurrence of excessive adjustment flow is reduced due to accumulation and superposition.

In the case of multi-canal pools requiring more water, the canal pools still operate according to the constant water level in front of the gate. The flow changes of most of the downstream canal pools are as follows:

$$\Delta Q(N,\ t) = \Delta q(N,\ t + \Delta \tau_N) + Q(t + \Delta \tau_N) \tag{13}$$

In the formula, $\Delta Q(N,\ t)$ represents the flow change of the $N$th canal and pond at time $t$, and $\Delta q(N,\ t - \Delta \tau_N)$ represents the water flow change of the $N$th canal and pond at time $t - \Delta \tau_N$. $Q(t - \Delta \tau_N)$ represents the $N$th channel pool recalling the flow at time $t - \Delta \tau_N$. $\Delta \tau_N$ represents the lag time of the $N$th channel pool.

For $N - 1$ channel pools:

$$\Delta Q(N-1,\ t) = \Delta q(N-1,\ t + \Delta \tau_{N-1}) + \Delta q(N,\ t + \Delta \tau_N + \Delta \tau_{N-1}) + Q(t + \Delta \tau_N) + Q(t + \Delta \tau_{N-1}) \tag{14}$$

and so on, until the first canal pool:

$$\Delta Q(1,\ t) = \Delta q(1,\ t + \Delta \tau_1) + \Delta q(2,\ t + \Delta \tau_1 + \Delta \tau_2) + \ldots + \Delta q(N,\ t + \Delta \tau_1 + \Delta \tau_2 + \ldots + \Delta \tau_N) \\ + Q(t + \Delta \tau_1) + Q(t + \Delta \tau_2) + \ldots + Q(t + \Delta \tau_N) \tag{15}$$

The summing expression is as follows:

$$\Delta Q(i,\ t) = \sum_{A=i}^{N} \Delta q\left(A,\ t + \sum_{B=i}^{A} \Delta \tau_B\right) + \sum_{J=i}^{N} Q(t + \Delta \tau_J) \tag{16}$$

　　The above formula can be used to obtain the compensation strategy for each section of the canal and pond. The gate regulation period $T_m$ is selected as 2 h. With the water level constraint $\Delta e$, the gate overcurrent formula is used to inversely calculate the over-sluice flow $\Delta Q_s$ for the safe variation of the water level. For details, the reader is referred to Equation (4). According to Equation (17), the flow that needs to be regulated by the gate can be calculated, the amount of gate regulation can be calculated, and the safety regulation can be completed.

### 3.2. Improved PID Control Algorithm

3.2.1. Traditional PID Control Algorithm

　　In the canal system control, the downstream constant water level control method takes the stability of the downstream water level as the control objective and takes the deviation between the water level value $c(t)$ and the target value $r(t)$ as the input of the PID controller:

$$e(t) = r(t) - c(t) \tag{17}$$

　　Linear combinations constitute the standard equation of the control quantity:

$$u(k) = K_p \left[ e(t) + \frac{1}{T_I} \int_0^t e(t)dt + \frac{T_d de(t)}{dt} \right] \tag{18}$$

　　In the equation, $K_p$ is the proportional coefficient, $T_1$ is the integral time constant, and $T_d$ is the differential time constant.

　　In the PID controller, P (proportional link) is mainly used to reduce the water level deviation, I (integral link) is mainly used to reduce the error of the gate water level control, and D (differential link) mainly reflects the changing trend of the water level deviation. When the action changes, a correction amount is introduced to speed up the gate control action and reduce the adjustment time.

　　Since the computer control system needs to discretize the standard equation of the control variable, a time T is selected to discretize the standard equation, and this time T is consistent with the discrete time interval of the Saint-Venant equations. The discrete PID control equation can be obtained by replacing the integral with the sum equation and the differential with the increment, as follows:

$$u(t) = K_p e(k) + K_I \sum_{j=0}^{k} e(j) + K_d [e(k) - e(k-1)] \tag{19}$$

　　In the equation, $k$ is the time series number, $k = 0, 1, 2, \ldots$, $u(t)$ is the output value of the controller at the $k$th time, that is, the increment of the gate, $e(k)$ is the deviation value of the water level at the kth time, $K_1$ is the integral coefficient, and $K_d$ is the differential coefficient.

3.2.2. Improved PID Control Algorithm

　　In the conventional channel-water conveyance project, the three parameters $K_p$, $K_I$, and $K_d$ of the traditional PID control are all fixed, which greatly reduces the flexibility of the control algorithm in the actual project and severely reduces its accuracy. Therefore, in order to overcome the problem that the traditional PID control algorithm cannot accurately control the non-linear canal pool, a non-linear PID control algorithm is proposed. This improved control algorithm can correct the $K_p$, $K_I$, and $K_d$ in real time according to the information of the water level change. The principle is shown in Figure 6.

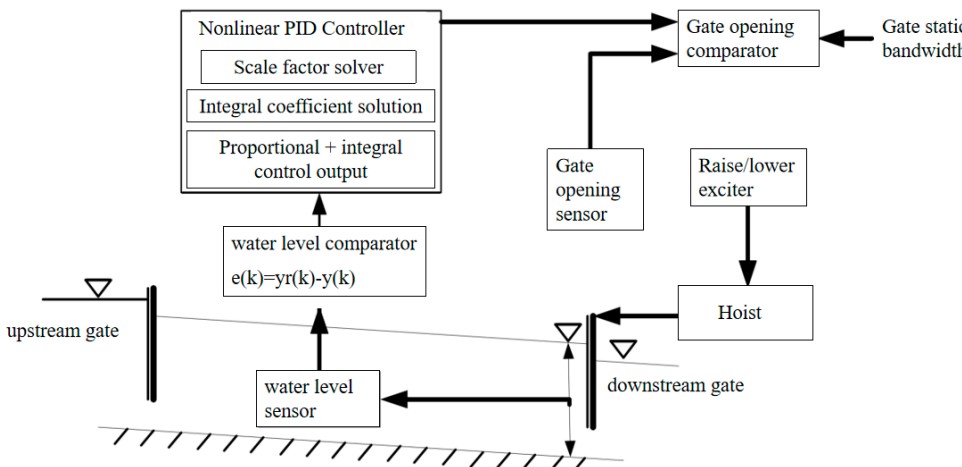

**Figure 6.** Schematic diagram of nonlinear PID control algorithm.

According to references [28–30], each gain parameter $K_p$, $K_I$, $K_d$ can be used to construct the following nonlinear function according to the water level change relationship:

$$K_p\big(e_p(t)\big) = a_p + b_p\big(1 - sech\big(c_p e_p(t)\big)\big) \tag{20}$$

$$K_I\big(e_p(t)\big) = a_I + b_I sech\big(c_I e_I(t)\big) \tag{21}$$

$$K_D\big(e_p(t)\big) = a_D + b_D\big\{1 + c_D exp\big[d_D sign\big(e_v(t)e_p(t)\big)\big]\big\} \tag{22}$$

In the Equation, $a_p$, $b_p$, $c_p$, $a_I$, $b_I$, $c_I$, $a_D$, $b_D$, $c_D$, and $d_D$ are positive real constants, and $e_v = e'_p$ is the error rate of change.

### 3.3. Storage Compensation Algorithm and Non-Linear PID Control Algorithm

The non-linear PID control algorithm is a closed-loop system that can ensure the stability of the system. However, sometimes, the fast response of the system is poor and there are oscillations; so, when the water consumption plan or water consumption flow is known, the feed-forward link is added in advance to enhance the fast response of the system. For constant water level operation in the downstream area, it is necessary to analyze and design the corresponding controller for the variation of the reservoir storage [31].

The sum algorithm in Figure 7 is as follows:

$$\Delta u = \begin{cases} sign(\Delta u_1)\max(|\Delta u_1|, |\Delta u_2|) \\ \Delta u_1 + \Delta u_2 \end{cases} \tag{23}$$

$$u = u_0 + \Delta u \tag{24}$$

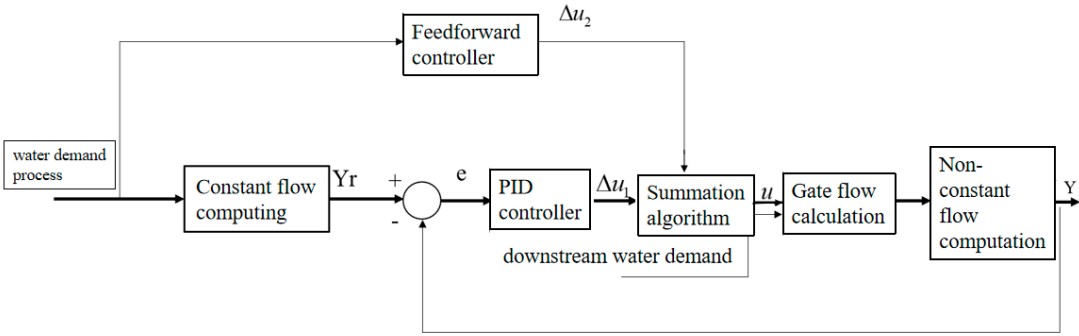

**Figure 7.** Schematic diagram of feedforward and feedback control.

## 4. Conclusions

In this research, a case channel system, four channels, and ponds after the middle line of the South-to-North Water transfer project are selected as the focus of this study. The water level change process of the channel pool system is studied for a large flow and a water distribution disturbance, the impact of the mouth flow change on the channel running water level is quantitatively analyzed, and a reference is provided for the dispatching and control of the open channel water transfer project. In this study, the bleeder is positioned downstream of the canal pool, and the constructed hydraulic simulation model is used to calculate the running bleeder conditions and summarize the effects of large-flow water bleeder changes on the running bleeder changes.

In this study, the middle pipe bleeder water separation flow is selected as the channel disturbance object. The flow rate changed from 0 m$^3$/s to 20 m$^3$/s in two hours (Figure 8). A tube head of the bleeder suddenly occurs after 24 h of the water disturbance event. The traditional storage quantity compensation algorithm and the coupling storage amount of compensation of the non-linear PID control algorithm of the two different algorithms respond to the canal pool overall water level disturbance and the middle pipe head bleeder from the large flow bleeder water distribution.

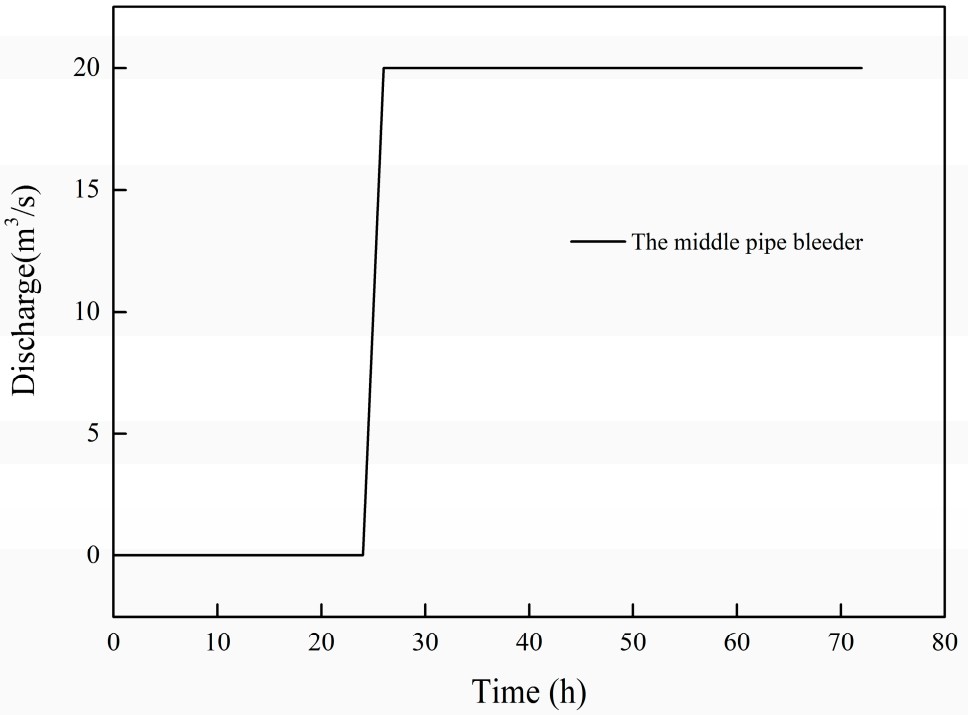

**Figure 8.** Downstream water demand process.

The control algorithm is used to establish the hydrodynamic model of the four channels after the middle route of the South-to-North Water Transfer Project. The upstream area of the channel pool selected by the model is assumed to be a fixed flow boundary process with a flow rate of 43 m$^3$/s, and the upstream area has a fixed water depth boundary with a water depth of 3 m. The duration of the hydrodynamic simulation is 3 days, and the calculated time step is 3600 s. The boundary settings are shown in Figure 9.

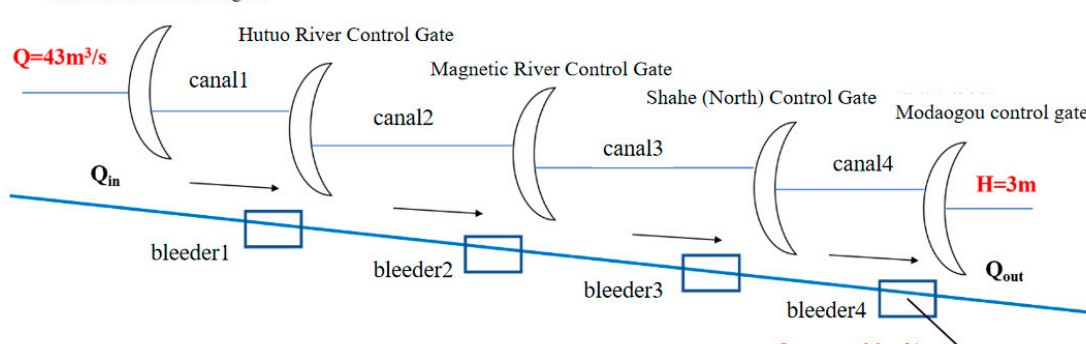

**Figure 9.** Boundary setting diagram.

In the hydraulic model, the upstream boundary flow, the downstream water depth, the length of each channel pool, the water flow rate of each channel pool, and the roughness of the channel pool are all parameters required by the model. The roughness of the channel pool is obtained after the model calibration, and the accuracy of roughness directly affects the accuracy of the model simulation. Refer to Table 1 for specific model parameters required by this model. The flow rate and water depth in Table 1, respectively, refer to the flow rate of the downstream control gate and the water depth before the downstream gate of the canal pool.

**Table 1.** Canal pool parameters.

| Channel Number | Gate | | Length (km) | Discharge (m³/s) | Water Depth (m) | Outflow (m³/s) | Roughness | Canal Bottom Elevation | |
| | Up | Down | | | | | | Up (m) | Down (m) |
| --- | --- | --- | --- | --- | --- | --- | --- | --- | --- |
| 1 | Guyunhe gate | Hutuohe Gate | 9.87 | 43 | 3.3 | 0 | 0.023 | 70.40 | 69.86 |
| 2 | Hutuohe Gate | Cihe Gate | 22.03 | 43 | 3.21 | 0 | 0.023 | 69.86 | 69.13 |
| 3 | Cihe Gate | Shahebei Gate | 15.21 | 43 | 3.03 | 0 | 0.023 | 69.13 | 68.36 |
| 4 | Shahebei Gate | Modaogou Gate | 19.73 | 13 | 3 | 20 | 0.023 | 68.36 | 66.70 |

*4.1. Analysis of Control Results of Traditional Storage Compensation Algorithm*

This section introduces the use of a traditional storage compensation algorithm for the large-flow bleeder and the middle pipe bleeder for known water bleeder changes from 0 m³/s to 20 m³/s over a 2 h period in the 24 h. To ensure the stability of the water level in front of the sluice when water diversion occurs, the channel pool can be regulated in advance with the method of storage volume compensation.

For the case of multiple channels in series, the storage compensation time of each channel pool is calculated, and then, the feedforward time of each control gate is calculated using stacking. Through the storage compensation algorithm, the change in the water level in front of the control gate is plotted, as shown in Figure 10. The change of the opening of the control gate is shown in Figure 11.

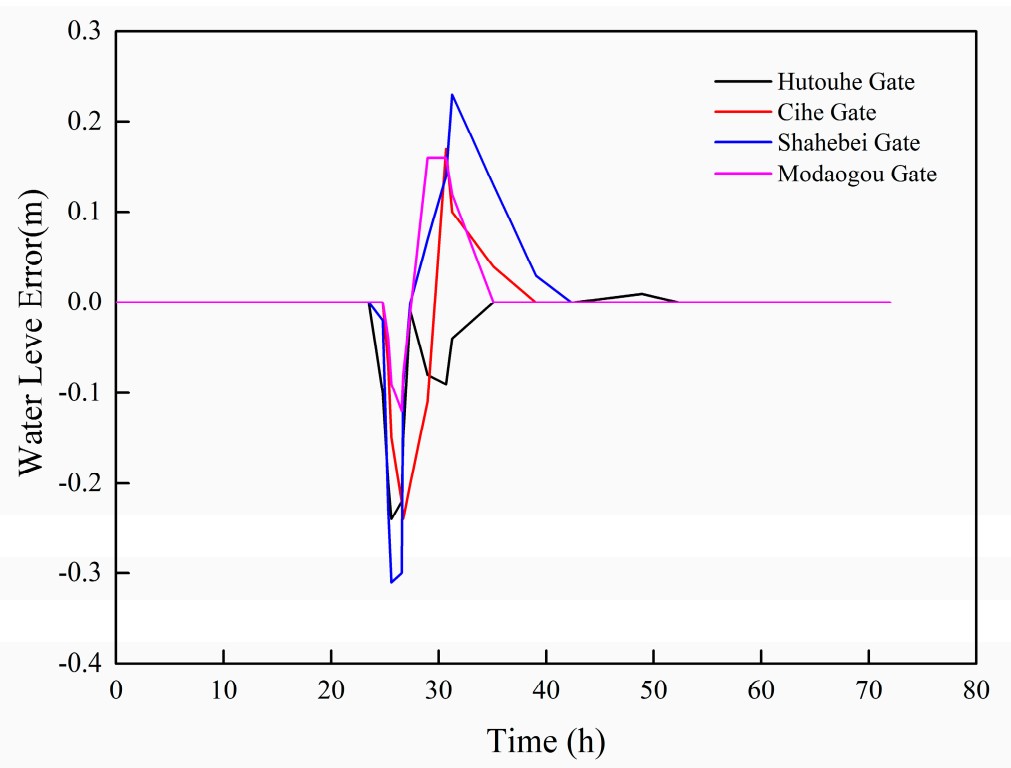

**Figure 10.** Variations of water level in front of the downstream sluice gate with the storage.

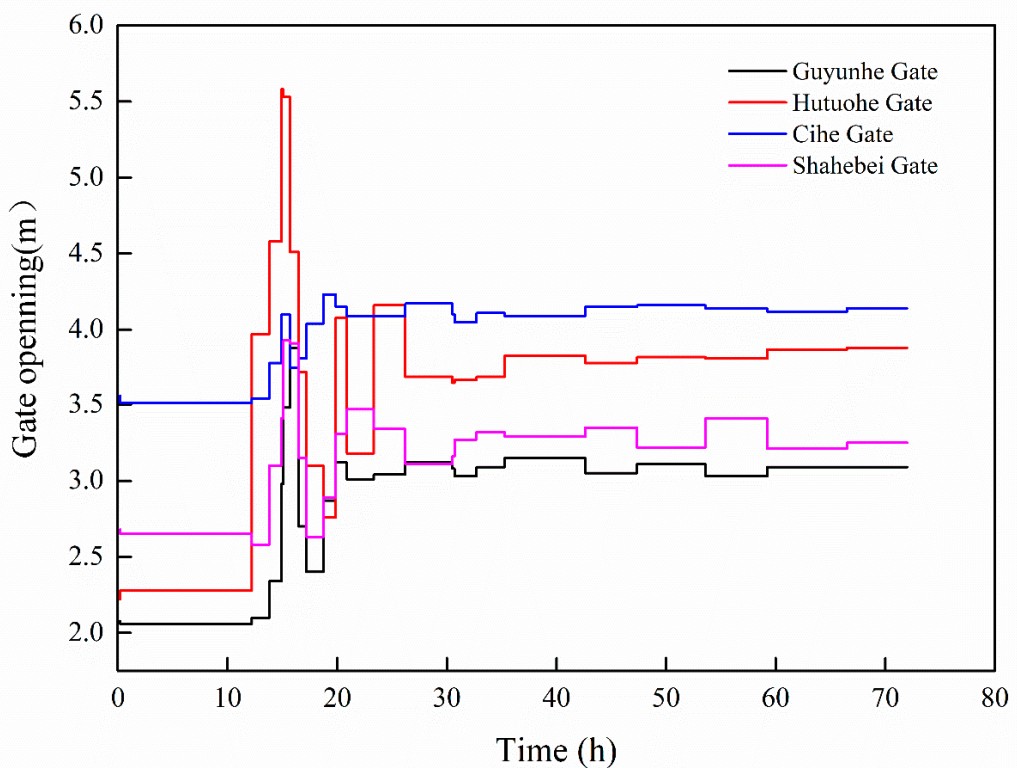

**Figure 11.** Process diagram of change of control gate opening upstream of canal pool.

As can be seen from Figure 10, in the case of 20 water discharge adjustments in the middle pipe water distribution, the water level in front of all the series control sluices is affected due to the strong coupling of the series canal pool in the middle line. The middle pipe bleeder is located in the Shahebei Gate–Modaogou Gate control sluice section. It can

be seen from Figure 10 that although the water level is finally stabilized in the target water level of the constant water level in front of the gate, the fluctuation phenomenon still occurs in the whole control process. Due to the 20 flow rate changes in the middle pipe head, all the upstream control gate inlet flows are adjusted by 15 flow rates according to the concept of flow balance. According to the adjustment results, although the stability of the water level can be eventually maintained through simple feedforward control, the maximum variation amplitude of the water level in the whole regulation process is 0.3 m, which is far larger than the requirement of the middle line channel pool that the variation amplitude of the water level per hour should not be greater than 0.15 m; so, the feedforward control cannot meet the situational requirements of a large flow water distribution disturbance in the middle line. Therefore, it is necessary to introduce feedback regulation in the conventional water transfer operation.

*4.2. Analysis of the Results of the Storage Compensation and Improved PID Coupling Control Algorithm*

The traditional storage compensation model is coupled with the feedback regulation algorithm, which is conducive to the occurrence of excessive fluctuation in the water level in the dispatching of water to avoid the abnormal fluctuation of the water level caused by the water distribution disturbance, which endangers the operation safety of the canal pool. This paper describes the improvement of the traditional PID control algorithm using the particle swarm optimization algorithm for the tuning of the PID control coefficient. This reduces the calculation time of the tuning coefficient, and the optimal solution of the three coefficients can be obtained using the optimization algorithm of the objective function.

With the feedforward control coupled feedback control algorithm, the changes in the water level and upstream control gate opening under the same working conditions are dealt with, as shown in Figures 12 and 13.

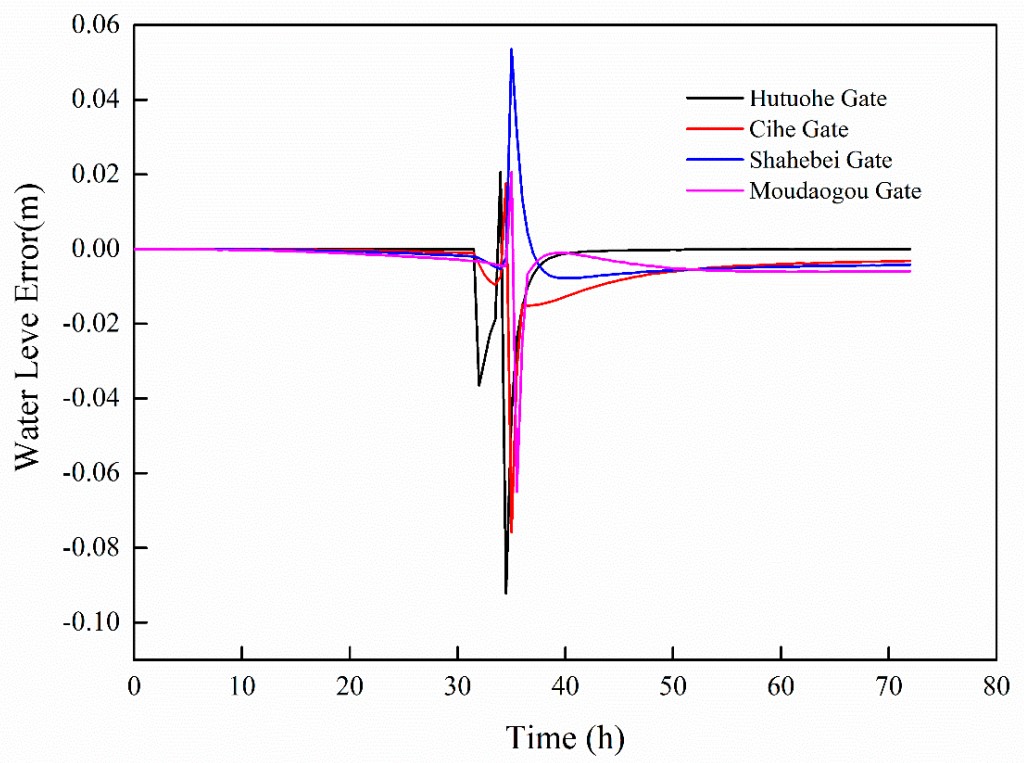

**Figure 12.** Water level change in front of the gate with the coupling control algorithm.

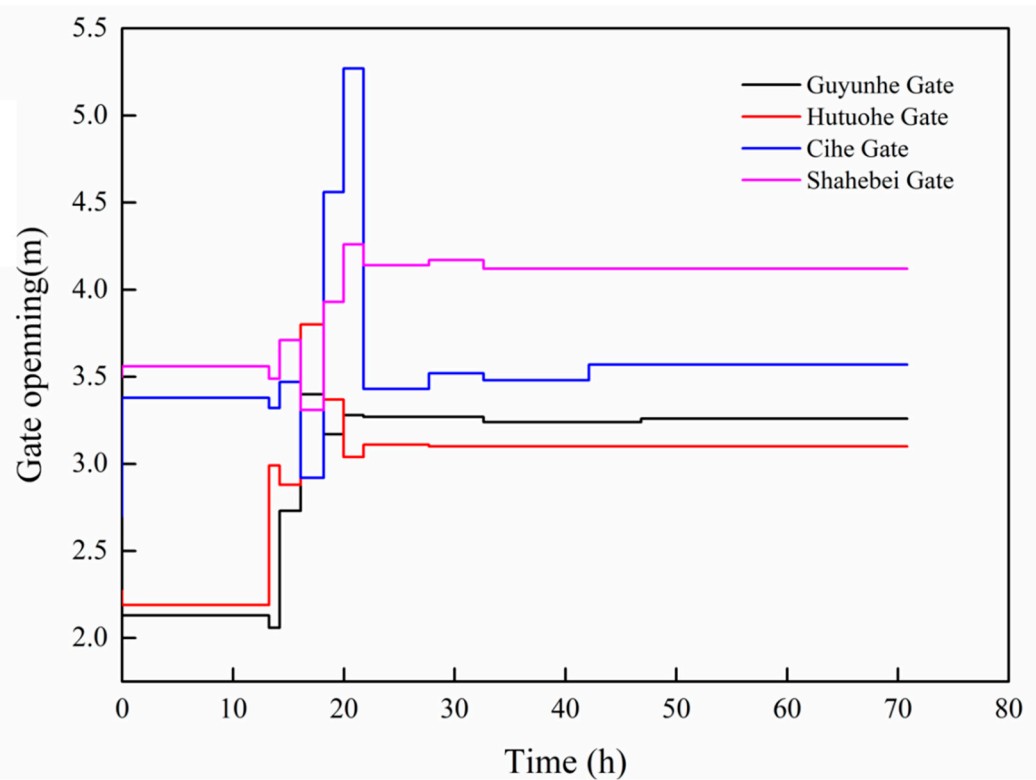

**Figure 13.** Coupling control algorithm upstream control gate opening variation diagram.

For the improved PID control algorithm, particle swarm optimization (PSO) is used to optimize the coefficients, and the $K_p$, $K_I$, and $K_d$ can be obtained by coupling the hydrodynamic model, as 0.32, 0.62, and 0.35, respectively.

Figure 12 shows the comparison of the coupling feedforward and feedback control algorithm model and the traditional storage quantity compensation algorithm, which has a great advantage for water level constraints. For the foreseeable water situation, this model can improve the charging amount compensation algorithm to adjust and control the gate to achieve a rapid response to the water level changes. However, due to the adjustment, the flow rate is larger, and due to the strong coupling series drainage pool, there is a cumulative effect variation range of the water level; so, further upstream, there is a relatively large variation range of the water level. The drainage pool has the greatest upstream water level fluctuation. To reduce the water level range, the PID control algorithm is introduced. The PID control algorithm is a condition triggering mechanism. When the water level exceeds the limit, the PID adjustment is started to ensure that the water level is within a proper range and to achieve the effect of canal pool water level stability.

It can be seen that the coupled control algorithm has a good control effect. Although it has a certain hysteresis, compared with the traditional control rules, the coupled control algorithm can reduce the control time. Therefore, the coupled control model can respond quickly to the water separation disturbance. The hysteresis of the PID control algorithm leads to the possibility of overshoot, which can be seen in the amplitude of the Ci River's water level.

It can be seen from Figure 12 that the maximum variation range of the water level is about 0.08 m, which is far less than the 0.15 m/h water level constraint condition of the middle line canal pool. Therefore, compared with the storage compensation algorithm, the coupled control algorithm can ensure the stable fluctuation of the water level of the canal pool under the conditions of large flow and a water distribution disturbance and can protect the operation safety of the canal pool.

**Author Contributions:** Writing—original draft preparation, J.S., B.K. and F.L.; writing—review and editing, J.S. and B.K.; supervision, Y.T. and X.S. All authors have read and agreed to the published version of the manuscript.

**Funding:** This research is financially supported by National Natural Science Foundation of China (Grant No. 42107162), Natural Science Foundation of Anhui Province (Grant No. 1908085QD168). The Fundamental Research Funds for the Central Universities of China (grant number: PA2021 KCPY0055).

**Data Availability Statement:** The data presented in this study are available on request from the corresponding author.

**Conflicts of Interest:** The authors declare no conflict of interest. The funders had no role in the design of the study; in the collection, analyses, or interpretation of data; in the writing of the manuscript, or in the decision to publish the results.

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
