# Peer review of "Study of a Control Algorithm with the Disturbance of Massive Discharge on an Open Channel"

_water, doi:10.3390/w14203252_

Round 1

Reviewer 1 Report

The topic of the article is interesting. Here you can find my detailed comments:

OVERALL: - Please, enlarge and re-arrange all Figures and font sizes to guide the reader properly in all sections. All figures must be composed of HD images. It is mandatory to improve the scientific quality of the whole manuscript.

-        Please, pay attention to the JOURNAL TEMPLATE in all sections, including tables, references, captions, units, and Figures.

-        Please, insert labels to all axes, and, most importantly, do not move them directly from EXCEL, but you must edit them differently, to be clear for all kinds of readers. This is mandatory.

INTRODUCTION: Please, consider in the scientific background of your study the use of the most advanced hydrodynamic methods recently employed for analyzing and monitoring water resources and systems broadly speaking (i.e.,

Khan, M.A., Sharma, N., Lama, G.F.C., Hasan, M., Garg, R., Busico, G., Alharbi, R.S. 2022. Three-Dimensional Hole Size (3DHS) Approach for Water Flow Turbulence Analysis over Emerging Sand Bars: Flume-Scale Experiments. Water 14, 1889. https://doi.org/10.3390/w14121889.

Wu, J. L., Xiao, H., & Paterson, E. (2018). Physics-informed machine learning approach for augmenting turbulence models: A comprehensive framework. Physical Review Fluids3(7), 074602.

Lama, G.F.C., Errico, A., Pasquino, V., Mirzaei, S., Preti, F., Chirico, G.B. 2022. Velocity uncertainty quantification based on Riparian vegetation indices in open channels colonized by Phragmites australis. J. Ecohydraulics 7(1), 71–76. https://doi.org/10.1080/24705357.2021.1938255)

METHODS: Please, insert a Figure for each sub-section. This will improve the scientific quality of your study, as a great support to all the equations proposed here.

CONCLUSIONS: The conclusion must be re-arranged according to the revisions indicated by the reviewer.

I SUGGEST MAJOR REVISIONS

Reviewer 2 Report

The current research focusses on open change discharge disturbances. As open channel flow plays critical role for river and channel hydraulics. Hence, it is necessary to maintain the water level near the gates when large flow disturbances occur in an open channel. Based on the ancient canal of the south-to-north water transfer project, a desert gap level drainage pool is taken as the research objective using two control algorithms, a feedforward compensation algorithm and a feedforward + feed-back control algorithm. The key finding of this study shows that the storage compensation algorithm can inhibit the fluctuation of water level to a certain extent, but the control effect of the storage compensation algorithm is still not satisfactory for the large flow water distribution disturbance. However, in the case of a large flow disturbance, the model of coupled control algorithm can stabilize the water level at an appropriate interval through the feedforward + feedback regulation mechanism so that the water level fluctuates well in the case of disturbance and the operation safety of the canal pool is protected.

However, the results plotted and the tables are at this stage elementary. The scope of the study is wide enough but in present form, the manuscript needs several changes. The authors may improve the manuscript with more research contribution and discussions. Many key parameters have not been considered in this analysis. The main limits concern: the confusion and lack of details in the presentation; the incomplete literature review where many important works on the investigated topic are. I feel major revisions are necessary for finalization of your work. Please consider my comments as given below. However, I have major concerns regarding the current version of manuscript which are mentioned below.

      I.         Abstract: The abstract could be supported by some quantitative findings. Some of the main results (findings) must be added in the abstract part.

    II.         Introduction & Literature Review: Literature review should be strengthened.

  III.         Can author elaborate the numerical methodology in details?

  IV.         Conclusions: The section should be supported by quantitative findings.

    V.         In General: Language of the text is fine, could be polished in some sentences.

  VI.         Check out the details of the references cited.

VII.         For all Figures: fonts and style should be uniform in both the sections. Also, legends style and font type and size should be uniform and same. Formatting of the Figures should be consistent with each other’s.

VIII.         The notations and equations are improperly formatted. Please see the journal's guidelines.

  IX.         The author did not specify which parameters influence the scour process around the bridge pier. They immediately displayed such characteristics in tables 4 and 5. Explain them prior in order to avoid confusion among the readers.

    X.         Most important, some important and latest literature is missing from the manuscript. Several recently published research papers are available in this research area. 

Reviewer 3 Report

Having read the paper Study of a control algorithm with the disturbance of massive discharge on an open channel, I believe the research is interesting and suitable for publication. However, the presentation needs to be improved:

  • In eq. 1, please define Z. Is this the same as the lower case z mentioned on page 3?

  • In eq. 2 please define g and S_0. Furthermore, if eq. 2 is the momentum equation, it needs to have vector components. Is this the momentum along the direction of the flow? (I may have missed some variable that has not been defined. Please double check.)

  • On line 74, the authors mention a non-uniformity correction coefficient alpha, which does not appear in the equations.

  • Figs. 6-10, please put the variables and units on the axes.

  • Regarding the solution to the Saint-Venant equations, I miss references to the work done By Simeoni (2002) and Hayat & Shang (2018)

Round 2

Reviewer 1 Report

The article has been improved according to the suggestions of the reviewer. Few issues must be fixed:  

INTRODUCTION: Please, consider in the scientific background of your study the importance of uncertainty quantification also in the management of water resources broadly speaking (i.e.,

Lama, G.F.C., Errico, A., Pasquino, V., Mirzaei, S., Preti, F., Chirico, G.B. 2022. Velocity uncertainty quantification based on Riparian vegetation indices in open channels colonized by Phragmites australis. J. Ecohydraulics 7(1), 71–76. https://doi.org/10.1080/24705357.2021.1938255

Khan, M.A., Sharma, N., Lama, G.F.C., Hasan, M., Garg, R., Busico, G., Alharbi, R.S. 2022. Three-Dimensional Hole Size (3DHS) Approach for Water Flow Turbulence Analysis over Emerging Sand Bars: Flume-Scale Experiments. Water 14, 1889. https://doi.org/10.3390/w14121889).

These points can considerably improve the scientific sound of the article overall.

Author Response

Thank you very much for your comments. Many thanks to the experts for their advice. I have added a discussion on the uncertainty of open channel flow patterns in the introduction.

Reviewer 2 Report

Authors have made all the required corrections. Hence editor can accept the manuscript.

Author Response

Thanks very much to the experts for their recognition of my paper.
